# The Identification of New c-FLIP Inhibitors for Restoring Apoptosis in TRAIL-Resistant Cancer Cells

Katherine Yaacoub [1,2], Rémy Pedeux [2], Pierre Lafite [3], Ulrich Jarry [1], Samia Aci-Sèche [3], Pascal Bonnet [3], Richard Daniellou [3,†,‡] and Thierry Guillaudeux [1,2,*]

[1] CNRS, INSERM, BIOSIT UAR 3480, US-S018, Rennes University, F-35000 Rennes, France; kathy_yaacoub@hotmail.com (K.Y.); ulrich.jarry@univ-rennes.fr (U.J.)

[2] INSERM, OSS (Oncogenesis Stress Signaling), UMR-S1242, CLCC Eugène Marquis, Rennes University, F-35000 Rennes, France; remy.pedeux@univ-rennes1.fr

[3] CNRS, ICOA, UMR 7311, Orléans University, F-45067 Orléans, France; pierre.lafite@univ-orleans.fr (P.L.); samia.aci@cnrs-orleans.fr (S.A.-S.); pascal.bonnet@univ-orleans.fr (P.B.); richard.daniellou@agroparistech.fr (R.D.)

[*] Correspondence: thierry.guillaudeux@univ-rennes1.fr or tguillaudeux@kineta.us

[†] Present address: Cosmétologie, AgroParisTech, 10 Rue Léonard de Vinci, F-45100 Orléans, France.

[‡] Present address: INRAE, AgroParisTech, UMR Micalis, Paris-Saclay University, F-78350 Jouy-en-Josas, France.

**Abstract:** The catalytically inactive caspase-8-homologous protein, c-FLIP, is a potent antiapoptotic protein highly expressed in various types of cancers. c-FLIP competes with caspase-8 for binding to the adaptor protein FADD (Fas-Associated Death Domain) following death receptors' (DRs) activation via the ligands of the TNF-R family. As a consequence, the extrinsic apoptotic signaling pathway involving DRs is inhibited. The inhibition of c-FLIP activity in tumor cells might enhance DR-mediated apoptosis and overcome immune and anticancer drug resistance. Based on an in silico approach, the aim of this work was to identify new small inhibitory molecules able to bind selectively to c-FLIP and block its anti-apoptotic activity. Using a homology 3D model of c-FLIP, an in silico screening of 1880 compounds from the NCI database (National Cancer Institute) was performed. Nine molecules were selected for in vitro assays, based on their binding affinity to c-FLIP and their high selectivity compared to caspase-8. These molecules selectively bind to the Death Effector Domain 2 (DED2) of c-FLIP. We have tested in vitro the inhibitory effect of these nine molecules using the human lung cancer cell line H1703, overexpressing c-FLIP. Our results showed that six of these newly identified compounds efficiently prevent FADD/c-FLIP interactions in a molecular pull-down assay, as well as in a DISC immunoprecipitation assay. The overexpression of c-FLIP in H1703 prevents TRAIL-mediated apoptosis; however, a combination of TRAIL with these selected molecules significantly restored TRAIL-induced cell death by rescuing caspase cleavage and activation. Altogether, our findings indicate that new inhibitory chemical molecules efficiently prevent c-FLIP recruitment into the DISC complex, thus restoring the caspase-8-dependent apoptotic cascade. These results pave the way to design new c-FLIP inhibitory molecules that may serve as anticancer agents in tumors overexpressing c-FLIP.

**Keywords:** c-FLIP; TRAIL; apoptosis; protein–protein interaction; drug resistance; cancer treatment

## 1. Introduction

The development of drug resistance is one of several clinical challenges causing anticancer drugs' limited efficacy or failure. The identification of resistance mechanisms to therapies helps to design novel therapeutics [1,2]. Over the past ten years, evidence has shown that abnormalities in apoptosis signaling pathways, such as the activation of anti-apoptotic proteins, were highly associated with immune and drug resistance [3].

Among these proteins, c-FLIP (Cellular FLICE Inhibitory Protein) is a major anti-apoptotic and resistance protein, restraining apoptosis induced by the TNF (Tumor Necrosis

Factor) superfamily members, including TRAIL (TNF-Related Apoptosis Inducing Ligand), Fas-Ligand or TNFα, as well as apoptosis stimulated by chemotherapeutic drugs in cancer cells [4]. c-FLIP is overexpressed in many different types of human cancers such as ovarian carcinomas [5], colorectal carcinomas [6], gastric adenocarcinomas [7] and prostate carcinomas [8]. The upregulation of c-FLIP has also been detected in primary tissues from patients with lung adenocarcinomas [9], hepatocellular carcinomas [10], melanomas [11], B-cell chronic lymphocytic leukemia [12] and Hodgkin's lymphomas [13].

c-FLIP has thirteen distinct variants, but only three of them are expressed as proteins in human cells. These are known as c-FLIP (L), c-FLIP (s) and c-FLIP(R) [14]. The long form c-FLIP (L) with a 55 kDa molecular weight (MW) is similar to procaspase-8, containing two *N*-terminal tandem "Death effector domains" = DED1 and DED2, and a *C*-terminal caspase-like domain, lacking the catalytic cysteine residue responsible for the proteolytic activity of caspases. The short form c-FLIP (s) (26 kDa MW) is composed only of two DEDs (DED1 and DED2), without a caspase-like domain and a short *C*-terminus [15]. Another short form of the protein called c-FLIP (R) (27 kDa) is specifically expressed in a number of T and B cells such as Raji cells as well as in human primary T cells. It also contains *N*-terminal DEDs (DED1 and DED2), but with a short *C*-terminus composed of a stretch of residues playing a key role in the ubiquitination of c-FLIP [15,16]

c-FLIP is considered a key inhibitory molecule in the extrinsic apoptotic signaling pathway, preventing the homodimerization and autoactivation of procaspase-8/10, the initiator caspases of apoptosis. This extrinsic pathway, also called death receptor pathway, is induced by different ligands of the TNF superfamily (TRAIL, Fas-L, TNFα), binding to their respective death receptors (DRs) respectively (TRAIL-R1/R2, Fas, TNF receptor). The binding of these ligands induces the trimerization of the respective DRs, which in turn recruit the adaptor protein FADD (Fas-Associated Death Domain) [17]. Once FADD is recruited to the DRs, procaspase-8/10 binds by its DED2 to the DED domain of FADD, leading to DISC (Death Inducing Signaling Complex) formation and the activation of downstream caspases and subsequent apoptosis [18]. However, this apoptotic signaling pathway can be attenuated or totally inhibited by c-FLIP. First, it was suggested that c-FLIP binds by its DED2 to the DED of the FADD and impedes the recruitment of procaspase-8 to the DISC, thereby precluding its activation [19]. Then, Scaffidi and coworkers contradicted this hypothesis, and demonstrated that caspase-8 is always recruited to the DISC at the same time as c-FLIP(s/L) proteins [20]. Procaspase-8 forms a heterodimeric complex with c-FLIP(L), resulting in an incomplete cleavage and limited activation of caspase-8, due to the lack of c-FLIP(L) enzymatic activity. This heterodimerization prevents further apoptotic signal transduction. By competition, c-FLIP(s) over-expression can also inhibit the processing of caspase-8 at the DISC, thus blocking the activation of the apoptotic cascade. These findings reflect the different functional roles of c-FLIP(L) and c-FLIP(s) in the inhibition of apoptosis [21].

TRAIL, also called APO-2L, is a member of TNF family and is mainly expressed by immune cells. It is a type II transmembrane protein with a C-terminal extracellular domain, which can be cleaved by a cysteine protease, resulting in a soluble form [22]. TRAIL can bind to five distinct receptors. TRAIL-R1 (DR4) and TRAIL-R2 (DR5) are classical death receptors and can trigger apoptosis with their functional cytoplasmic death domains (DDs). TRAIL-R3 (DcR1) and TRAIL-R4 (DcR2), also known as decoy receptors, as well the circulating receptor Osteoprotegerin (OPG) are not able to propagate a death signal due to a lack of a functional cytoplasmic death domain [23]. TRAIL expressed on cytotoxic T cells and NK cells is a potent anti-tumor molecule, since it has been proven to preferentially kill cancer cells in a wide variety of tumors and does not exhibit any toxicity in a majority of normal cells. However, a large number of cancers evade TRAIL-induced apoptosis and become TRAIL resistant through different mechanisms, including the over expression of c-FLIP [24]. The selective knock-down of c-FLIP(L) sensitizes tumor cells to TRAIL-induced cell death in human lung cancer cell lines [25]. It has also been demonstrated that Withanolide E, a steroidal lactone derived from Physalis peruviana, can highly sensitize

renal carcinoma cells and other human cancer cells to TRAIL-mediated apoptosis through the rapid destabilization, aggregation and proteasomal degradation of c-FLIP proteins, confirming the key inhibitory role of c-FLIP in death ligand-mediated apoptosis [26].

So far, except for siRNAs approaches, c-FLIP inhibitors that have been studied act indirectly on c-FLIP, such as Cisplatin, which induces p53-dependent FLIP ubiquitination and degradation in ovarian cancer cells [27], or Actinomycin D, which downregulates FLIP(L) and FLIP(s) expression in B chronic lymphocytic leukemia [28]. Thus, the identification of small molecules more stable than siRNA and directly targeting c-FLIP represents a new promising strategy to overcome therapy resistance. c-FLIP is structurally similar to caspase-8. Each DED of c-FLIP shares ~25% similarity with the DEDs of caspase-8, and the C-terminus (270 amino acids) of c-FLIP is also ~25% identical to the C-terminus of caspase-8 [29]. Thus, due to this homology, the identification of new compounds that selectively bind to c-FLIP versus caspase-8 and prevent its recruitment to the DISC is challenging.

Based on this rationale, molecular modeling and docking experiments were set up to construct c-FLIP and caspase-8 homology models, in order to find selective inhibitors targeting unique sequences of c-FLIP and not caspase-8. In vitro assays using recombinant c-FLIP(s) and FADDs, coupled with in cellulo assays, demonstrated the inhibitory role of these new molecules, restoring TRAIL-mediated apoptosis and selectively preventing c-FLIP/FADD interactions. Our findings suggest that blocking c-FLIP's recruitment into the DISC by specific inhibitors decreases tumor resistance to death receptor-mediated apoptosis, and represent a new avenue for cancer treatment.

## 2. Materials and Methods

### 2.1. Molecular Modeling

#### 2.1.1. The Homology Modeling of DED2s

Homology models were built using MOE software (Molecular Operating Environment, v2012.10). First, sequences of the DED2 domains of c-FLIP and of CASP8 were extracted from the Uniprot database and used to find homolog structures in the Protein Data Bank (PDB) using the BLAST software (https://www.uniprot.org/blast, accessed on 1 January 2013). Three sequences showed a significant percentage of identity with the target sequences (around 30%) to perform homology modeling. Three structures of the FADD DED domain (PDB.ID 1A1W, 1A1Z, 2GF5) and three structures of two v-FLIP DED2s (PDB.ID 2BBR, 2BBZ, 3CL3) were identified as suitable templates for modeling the DED2s of CASP8 and c-FLIP, respectively.

#### 2.1.2. The Identification of the Binding Site

SiteFinder module of the MOE software was used to identify druggable pockets at the surface of our models, and only the homology models which presented such pockets at the vicinity of the F-L hydrophobic patch were retained. Two models of the CASP8 DED2 and one model of the c-FLIP DED2 were kept based on the PDB.ID 1A1W, 2GF5 and 2BBZ structure templates, respectively.

#### 2.1.3. The Docking of Chemical Libraries

The 1880 molecules from the NCI DiversitySet3 and extracted from the ZINC database were virtually screened on the three models (two for CASP8 DED2 and one for c-FLIP DED2) using two kinds of docking software: AutoDock (version 4.2 with associated tool MGLTools version 1.5.6rc3) and Glide (program included in the Schrödinger software suite from release 2013). The ZINC database provides ready-to-dock sets of purchasable molecules. The results of these two virtual screenings were combined using a consensus scoring method and a root mean square deviation (RMSD) filter.

#### 2.1.4. Consensus Scoring Function

In addition to the Glide and AutoDock scoring functions, the MOE GBVI/WS dG function was selected to rescore all poses of the docked ligands. Therefore, the binding

modes of each docked ligand with Glide and AutoDock were assessed using four score values. The score values were normalized using the *Z-score* following formula:

$$Z\text{-}score = \frac{score - \mu}{\sigma}$$

where $\mu$ is the mean and $\sigma$ is the standard deviation of the scores.

The four normalized scores were then summed up and ranked by decreasing order, the best score being the lowest ones. To keep only ligands presenting similar poses with the two docking methods, we calculated the RMSD (Root Mean Square Deviation) between the poses obtained by the two methods. The ligands which presented a RMSD value higher than 2 Å were removed from the ranking.

### 2.1.5. Hit Selection

The goal of this docking study was to identify ligands that selectively bind to c-FLIP and not to CASP8. For this purpose, the best 20 screened molecules obtained on the c-FLIP target domain were compared with the best 20 molecules obtained on each CASP8 model. Molecules present in the top 20 for c-FLIP and absent in the top 20 for CASP8 models were considered selective ligands. Among these molecules, we selected only the three top molecules obtained by each docking software: AutoDock, Glide and consensus strategy. The final nine molecules were then used to test their activities in vitro.

### 2.2. Cell Culture

H1703 (Human non-small lung cancer cell line) and mock transfected or overexpressing c-FLIP(L) cells were grown in RPMI 1640 (LONZA) culture media, supplemented with 10% fetal bovine serum and puromycin (2 μg/mL) from Sigma-Aldrich. The cells were kept in a humidified atmosphere in an incubator at 37 °C and 5% $CO_2$. The H1703 was a kind gift from O. Micheau (Inserm, Dijon, France).

### 2.3. Reagents and Antibodies

KillerTRAIL (human recombinant) was purchased from Alexis Biochemicals. The nine most highly selective molecules targeting cFLIP were obtained from NCI–DSCB (National Cancer Institute-Drug Synthesis and Chemistry Branch, Rockville, MD, USA). For Western blotting (WB) experiments, we used the following antibodies: anti-FLIP antibodies DAVE2 and NF6 (Adipogen, San Diego, CA, USA), caspase-3 (8G10; OZYME), PARP (Asp214, 19F4; OZYME, Saint-Cyr-l'École, France), caspase-8 (1C12; OZYME), anti-MBP (New England Biolabs, Ipswich, MA, USA) and anti-His (C-ter) (Invitrogen, Carlsbad, CA, USA). Anti-His (ab81663) was used to immunoprecipitate the DISC complex, and the following antibodies were used for WB analysis: anti-Flip (DAVE II, Adipogen), anti-FADD (556402, BD, San Jose CA, USA), anti-casp8 (5F7, EnzoLife, Farmingdale, NY, USA), anti-DR4 (1139, ProSci, Poway, CA, USA) and anti-DR5 (3696, Cell Signaling, Danvers, MA, USA). Anti-rabbit and anti-mouse HRP-linked secondary antibodies (Santa Cruz Biotechnology, Dallas, TX, USA) and β-actin (Sigma Aldrich, St. Louis, MO, USA) were also used.

### 2.4. Flow Cytometry Analysis

Apoptotic cell death was confirmed by flow cytometry (cytoFLEX, Beckman Coulter, Brea, CA, USA) using Annexin V-PE Kit, according to the manufacturer's instructions (BD Biosciences, USA). In brief, H1703 cells were seeded at a density of $2 \times 10^5$ cells/well in 24-well plates and incubated for 24 h. Cells were then treated, as indicated in figure legends, for 18 h. Thereafter, cells were collected, washed and re-suspended in $1\times$ Annexin-V binding buffer. Annexin-V-PE was added to the cells and left for 20 min at room temperature in the dark. A 7-AAD dye was added and flow cytometric analysis was performed in the final step.

### 2.5. Recombinant Protein Production and Purification

Full-length c-FLIP(s) and FADD were synthesized (Genscript, Piscataway, NJ, USA) and subcloned, respectively, into pET24b(+) (Novagen, Madison, WI, USA) and pMAL-C2X (New England Biolabs) expression vectors. The resulting constructs enabled the fusion of the corresponding protein with a *C*-terminal poly-histidine peptide, or a *N*-terminal Maltose Binding Protein (MBP).

All proteins were expressed in 1 mM IPTG-induced Rosetta-transformed bacterial cells with the expression vectors. After 18 h of induction at 37 °C, cells were harvested and pellets were resuspended in a lysis buffer (50 mM Tris-HCl, 100 mM NaCl, pH 8, 0.1% Triton), in addition to 0.1 mg/mL lysozyme and 1 mM PMSF (Phenylmethanesulfonyl fluoride), and incubated for 20 min at 4 °C. Cells were then lysed by freeze–thaw cycles, followed by sonification. Lysate was centrifuged at $34,000\times g$ for 20 min at 4 °C, and supernatant was loaded on chromatographic media. HisPur$^{TM}$ Ni-NTA Chromatography Cartridge 1 mL (Thermo Scientific, Waltham, MA, USA) or MBPTrap$^{TM}$ HP 1 mL (GE Healthcare, Chicago, IL, USA) were, respectively, used for purification of c-FLIP(s) or FADD, following the column's manufacturer procedures. Protein purity was assessed by SDS-PAGE analysis and concentrations were qualified using the Bio-Rad protein assay, based on the Bradford dye-binding method.

### 2.6. Pull-Down Binding Assay

The purified proteins were mixed at a ratio of 1:7 for FADD:FLIP(s), equivalent to 0.7 mg/mL for FADD and 6.8 mg/mL for FLIP (26 μM and 261 μM, respectively), and incubated for 18 h at 4 °C with 0–3000 μM concentration range of each inhibitor. Then, the incubation mix was loaded on a MBPTrap chromatography cartridge and purified using 10 mM maltose as eluent. Negative controls were also performed without FADD. Unbound and eluted samples were used for Western Blot analysis.

### 2.7. DISC Immunoprecipitation

For DISC Immunoprecipitation, $50 \times 10^6$ H1703-FLIP(L) cells seeded in F175 Flasks were incubated overnight. The next day, cells were collected in 10 mL of media, pre-treated with selected Molecules 1, 3, 4 or 9 (500 μM) for 2 h and then stimulated with 1 μg/mL His-TRAIL for 20 min. Reaction was blocked with cold PBS, and cells were lysed in 1 mL of IP-Lysis Buffer (1% NP40, 20 mM TRIS HCL Ph 7.4, 150 mM NaCl, 10% Glycerol), containing COMPLETE Inhibitor of Proteases cocktail (Roche, Basel, Switzerland), for 30 min on ice. Lysates were cleared by centrifugation at 15,000 RPM, 20 min at 4 °C. All supernatants were pre-cleared with 50 μL of Sepharose 6B (Sigma 6B100) for 1.5 h at 4 °C on a wheel, and then centrifugated at 1500 RPM at 4 °C for 15 s. To analyze the DISC complex, supernatants were collected and incubated overnight at 4 °C with 50 μL Protein G beads and coupled with 3 μg anti-His tag antibody (Abcam ab81663, Cambridge, UK) supplemented with 1 μM of caspase inhibitor Z-VAD. After immunoprecipitation (IP), beads were washed 4 times with IP-lysis buffer (without inhibitor of proteases) and eluted with 120 μL of LDS buffer with DTT. Samples were left for 30 min at room temperature, then heated for 5 min at 95 °C before WB analysis.

### 2.8. Western Blot Analysis

Pull-down eluted samples were heated for 5 min at 99 °C with Laemmli Buffer and were then passed on 4–12% gradient SDS-PAGE and transferred to nitrocellulose membrane (GE Healthcare). To prepare cellular lysates, treated cells were lysed in RIPA buffer at 4 °C and centrifuged at 15,000 rpm for 20 min. Protein concentration was then determined using Bradford assay. Equal amounts of proteins (50 μg) were boiled for 5 min with 1X LDS sample buffer and then loaded on a 4–12% SDS-PAGE and transferred to a nitrocellulose membrane. Membranes were blocked by 5% non-fat milk in PBS-TWEEN 20 (0.1%) for one hour at room temperature and then incubated with different primary antibodies for one to two hours at room temperature. Membranes were then washed by PBS-TWEEN

and incubated for 1 h with HRP-conjugated secondary antibodies. Proteins' bands were visualized by chemoluminescence protocol ECL (Thermo Scientific).

*2.9. Statistical Analysis*

Statistical analyses were performed with the Student's *t*-test.

## 3. Results

*3.1. Potential c-FLIP Inhibitors Selected by In Silico Screening*

Three homology models were built for each target: the DED2s of CASP8 and c-FLIP. Among the six homology models obtained, only those with at least one druggable pocket near the hydrophobic patch F-L were conserved (Figure 1A). We thus kept one model for the c-FLIP DED2, based on the PDB.ID 2BBZ structure, and two models for the CASP8 DED2, based on the PDB.ID 1A1W and 2GF5 structures.

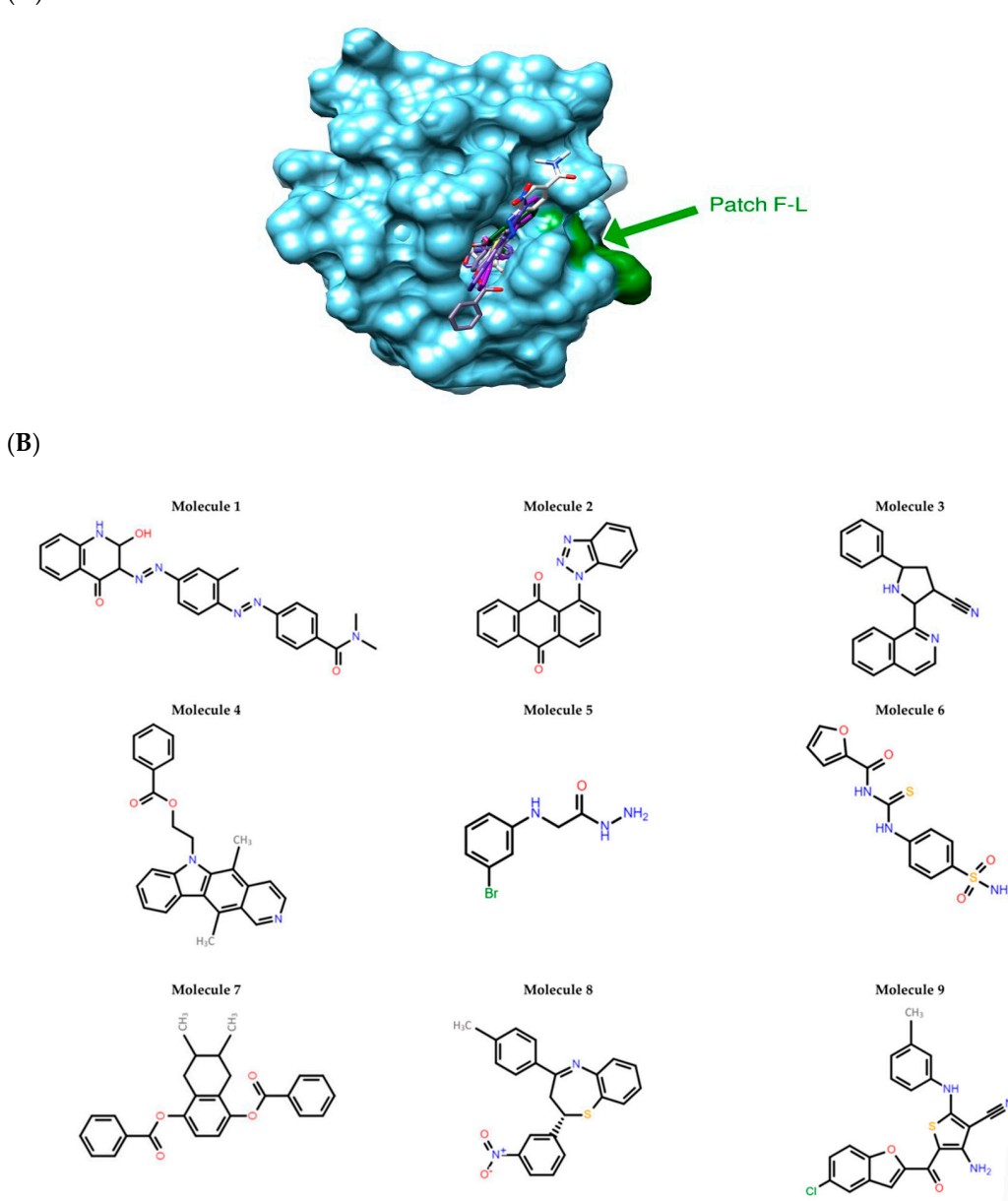

**Figure 1.** In silico identification of selective c-FLIP DED-binding molecules. (**A**) Best docking poses of the nine molecules in the binding pocket of the c-FLIP structural model. The binding pocket is a

"hydrophobic patch" of Phenylalanine-Leucine (F-L) motif which is highly conserved in DED domains. The SiteFinder module (MOE) allowed us to identify the druggable pockets. The molecules are in stick representation. c-FLIP structure model is represented as blue surface. The F-L patch is highlighted in green. (**B**) 2D structures of the nine molecules selected by in silico methods.

The search for druggable pockets near the hydrophobic patch allowed us to select the models to be used in virtual screenings: the CASP8 models based on 1A1W and 2GF5 (which we will, respectively, call CASP8 (1A1W) and CASP8 (2GF5)) and the c-FLIP model 2BBZ c-FLIP (2BBZ). The MOE SiteFinder module searches for cavities in a protein using a method based on filling with alpha centers. A deep cavity contains at least ten α centers. The pockets of the CASP8 (1A1W) model only contain seven, so they are rather small in surface. One of the pockets of the CASP8 (2GF5) model contains 21 α centers and the other less than ten. This model therefore has a surface pocket and a buried one. The c-FLIP (2BBZ) model has the largest and deepest cavity, as a tunnel-shaped (31 α centers). Pockets 1 and 2 of the CASP8 models are located on either side from the hydrophobic patch and always, respectively, present the triplets ARG19, LYS22 and PHE23 (pocket 1) and LEU24, VAL64 and GLN67 (pocket 2). The c-FLIP cavity is notably bordered by residues LEU24 and CYS64, and therefore corresponds to pocket 2 of the CASP8 models (Figure 2).

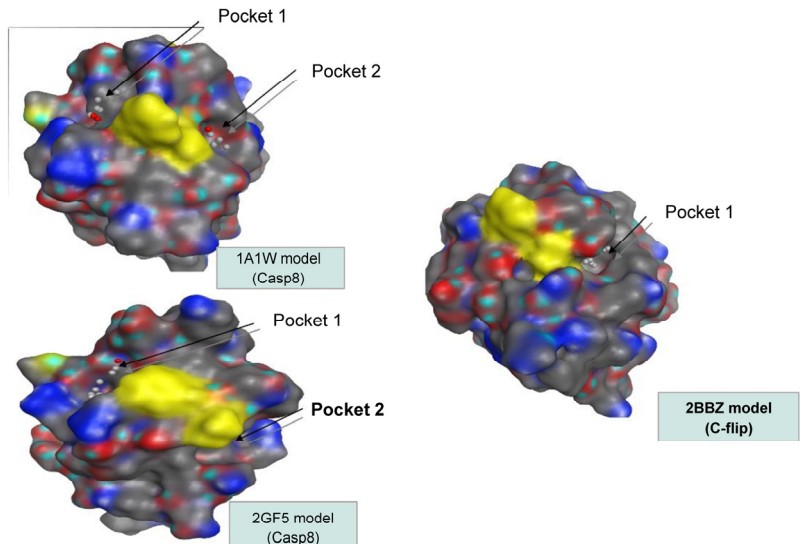

**Figure 2.** Representation of the CASP8 and c-FLIP models chosen to carry out molecular docking with locations of the pockets near the hydrophobic patch (yellow).

For each of the two models of the CASP8 DED2, two druggable pockets were found at the proximity of the hydrophobic patch F-L by the module SiteFinder of the MOE software. We only found one druggable pocket satisfying this condition for the homology model of the c-FLIP DED2. Five virtual screenings were performed using two docking Software: AutoDock and Glide, four on the CASP8 target and one on c-FLIP. We then combined the results obtained by the two docking software using a "consensus" method scoring, added with a filter based on the RMSD (Root-mean-square deviation). Consensus scoring allows to extract unique representative information from a set of results from different sources. Each docking software gives a different classification for different molecules binding on the druggable pockets. Those in the top of the classification are considered to have the highest binding affinity. Consequently, we selected the top three molecules from each of the three docking software (AutoDock, Glide, Consensus). These nine selected molecules were considered potential selective inhibitors of c-FLIP and were then further tested in biological assays (Figure 1B).

### 3.2. The Newly Identified c-FLIP Inhibitors Prevent DED–FADD/DED2–c-FLIP Interactions

Upon death receptors' stimulation and DISC formation, the DD domain of the death receptors interacts with the DD of the FADD, whereas the DED of the FADD interacts with the DED2 of procaspases 8/10 [18]. When highly expressed, c-FLIP is incorporated and binds to the FADD via its DED2 domain, particularly via the FL motif (F114/L115). Mutations in this FL motif abrogated FLIP recruitment into the DISC [30]. There are two contradictory pieces of evidence concerning c-FLIP binding to the FADD. One study has shown that the DED of the FADD is docked at the interface between the DED1 and DED2 of c-FLIP, because mutations in this linker region exhibited a weak interaction with the DED of the FADD [31]. However, another study has demonstrated that the amino acid F114 of the hydrophobic patch of the c-FLIP–DED2 is responsible for binding to the FADD's $\alpha$1-$\alpha$4 groove, with a reciprocal interaction of H9 of the FADD with $\alpha$–$\alpha$5 hydrophobic patch of DED2–c-FLIP [32]. However, the direct interaction between c-FLIP and the FADD DED is not very well determined. Here, to elucidate the molecular mechanism involved in the restoration of TRAIL-induced apoptosis by molecules 1 to 9, we investigated the potentiality of these compounds to prevent c-FLIP(s)/FADD interaction. A mix containing recombinant proteins c-FLIP(s) and FADD was incubated in the presence of various concentrations of each inhibitor, and the FADD was purified using the MBP affinity tag. When a potent inhibitor prevented the interaction, then c-FLIP(s) was not co-eluted after FADD purification.

As depicted in Figure 3A–D,I, molecules 1, 2, 3, 4, and 9 were able to prevent c-FLIP(s)/FADD direct interaction with a concentration below 3000 µM. This high concentration was required as c-FLIP(s) and the FADD have to be mixed at high concentrations to enable efficient pull-down in this assay. Molecules 5, 6, 7 and 8 (Figure 3E–H) did not exhibit a potent inhibition of the protein–protein interaction below the highest concentration tested. Interestingly, molecules 1 and 2 exhibited the highest potency, as they prevented the interaction with the lowest concentration (500 µM), followed by molecules 3 and 9 (1000 µM) and finally molecule 4 (2500 µM).

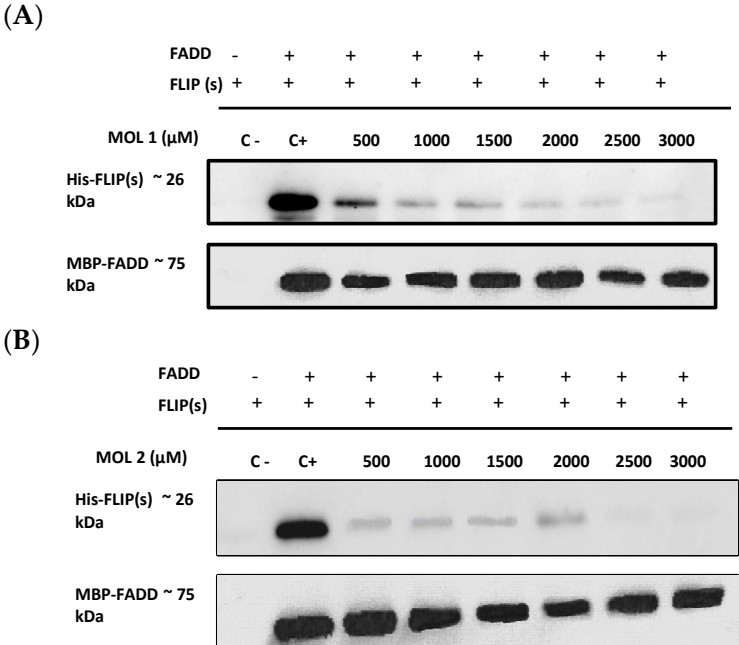

**Figure 3.** *Cont.*

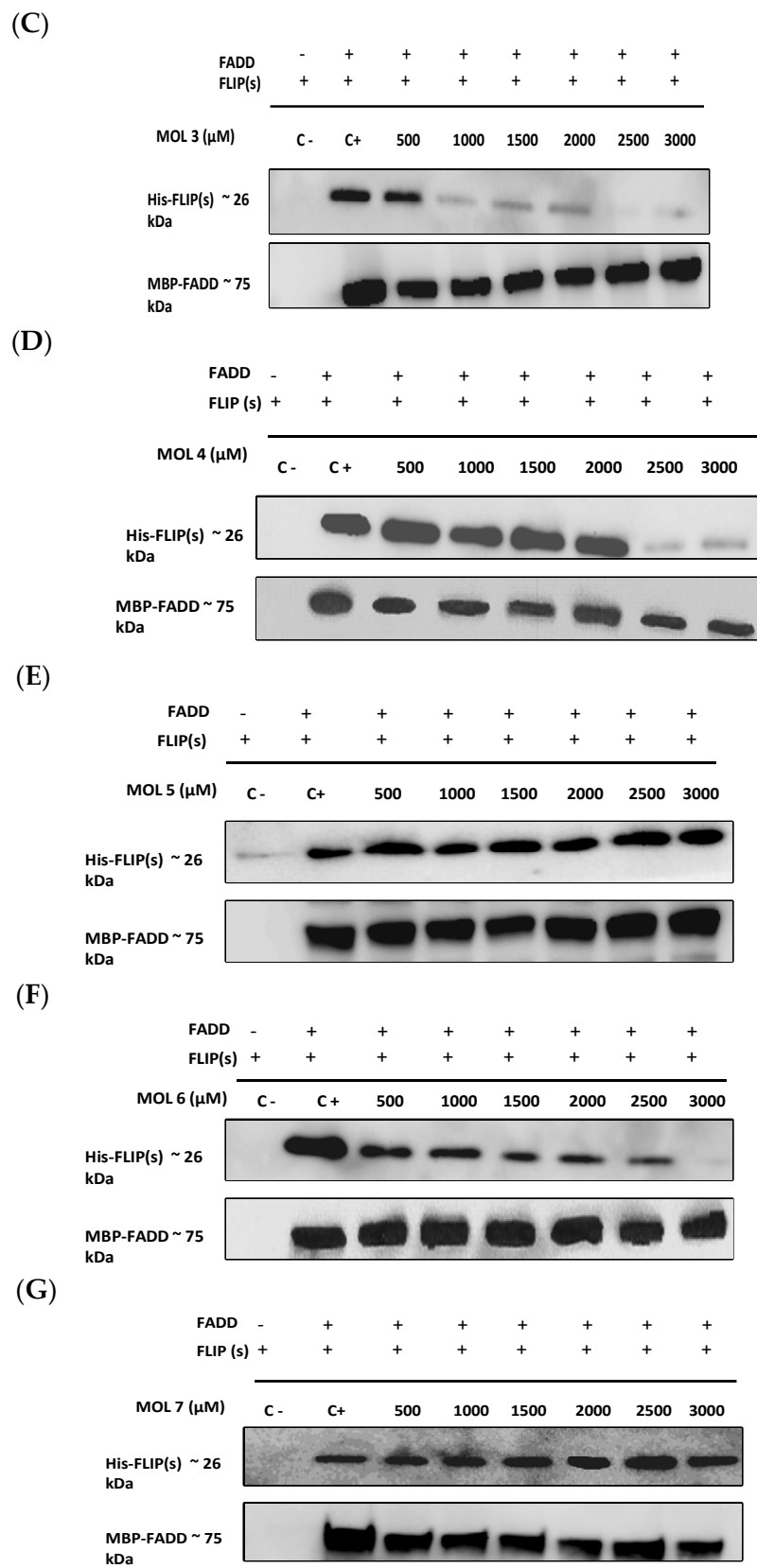

**Figure 3.** *Cont.*

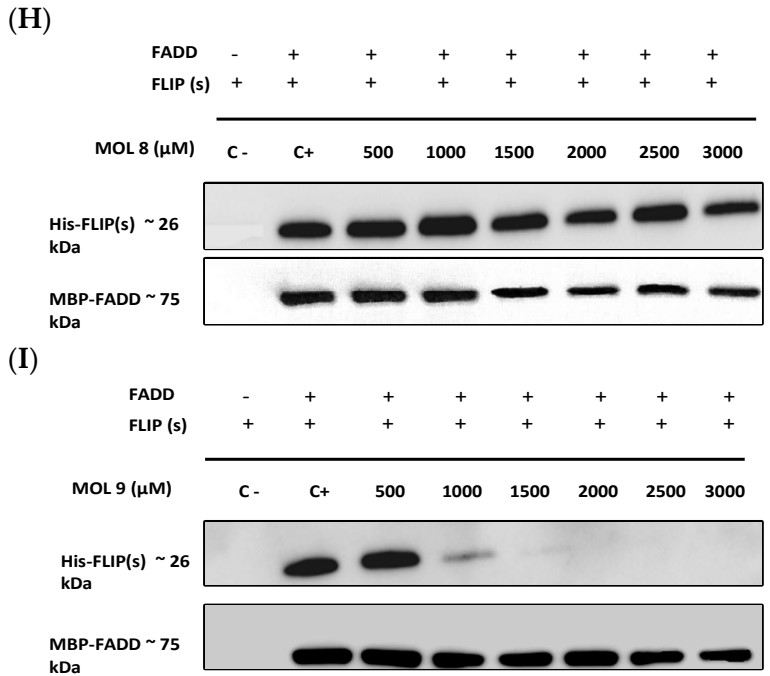

**Figure 3.** The newly identified inhibitors prevent DED–FADD/DED2–c-FLIP interaction (**A–I**). MBP-tagged FADD and His-tagged FLIP(s) were produced and purified using chromatography columns, and incubated with each inhibitor using the indicated range of concentrations, for 18 h. The different mixtures were passed on MBP-trap chromatography columns, and eluted samples were analyzed by Western Blot using antibodies directed against Histidine and MBP; C−: Negative control without FADD, C+: Positive control FADD and FLIP without any inhibitor.

### 3.3. Newly Identified Molecules Inhibit c-FLIP Recruitment into the DISC by Preventing Its Interaction with the FADD

It was important to confirm that our new selected molecules inhibit c-FLIP interaction with the FADD and its recruitment into the DISC, since it has been demonstrated that FLIP depletion from the cytosolic fraction prevents its binding to the DISC and allows for caspase-8 cleavage and apoptosis [33]. To evaluate the inhibitory role of these molecules, in cellulo tests were proceeded, using H1703 cells overexpressing the c-FLIP protein. The H1703 was treated with TRAIL, alone or in combination with FLIP inhibitors, then the DISC complex was immunoprecipitated to investigate the presence of the FLIP protein. Interestingly, when the cells were treated with TRAIL combined with Molecule 1, 3, 4 or 9, FLIP recruitment into the DISC was significantly reduced (Molecule 2 was not tested because of a lack of its required concentration). This was demonstrated by a significantly reduced elution of FLIP with other DISC components (Figure 4A,B). Remarkably, Molecule 3 and 9 seemed to be more efficient in preventing FLIP interaction with the DISC, according to the pull-down assay data (Figure 3). As a control, the FLIP protein was always recruited to the DISC when H1703 cells were treated with TRAIL alone. Hence, the new molecules act as inhibitors to prevent FLIP incorporation into the DISC complex following TRAIL stimulation.

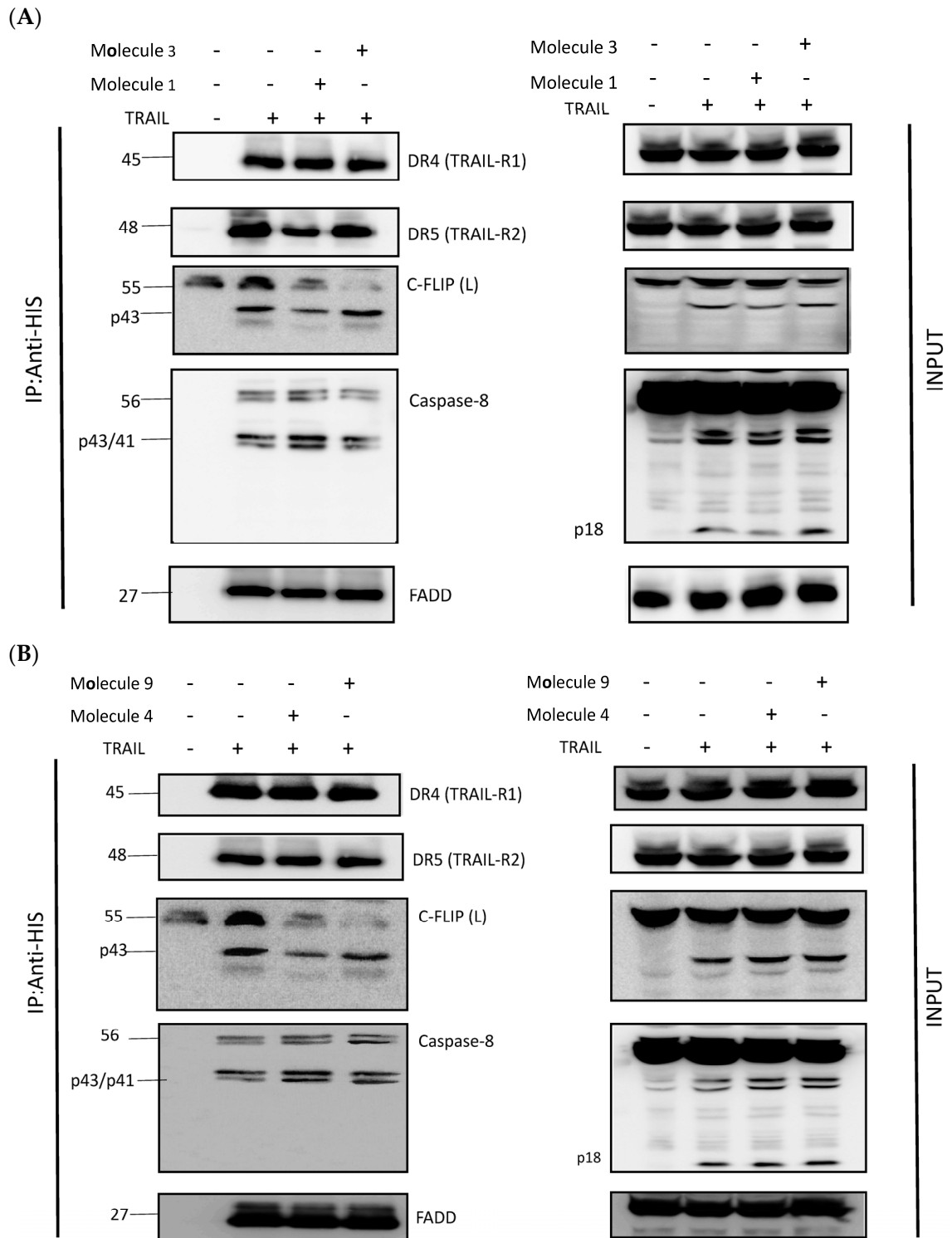

**Figure 4.** Newly identified molecules inhibit c-FLIP recruitment to the DISC complex and promote apoptosis. $50 \times 10^6$ H1703-FLIP(L) cells were seeded and exposed the next day to a "Pre-treatment" with 500 μM Molecule 1 and 3 (**A**), or Molecule 4 and 9 (**B**), for 2 h. Cells were treated later with 1 μg/mL of His-TRAIL for 20 min. DISC complexes were immunoprecipitated using anti-HIS antibodies and analyzed by Western Blot.

### 3.4. c-FLIP Inhibitors Rescue TRAIL-Mediated Apoptosis

It was reported previously that the upregulated expression of c-FLIP(L) precludes the interaction of the initiator procaspase 8/10 with the adaptor protein FADD, thereby blocking

cell death induced by members of the TNF superfamily (TNF-α, FasL, or TRAIL) [34]. Indeed, our previous study revealed that TRAIL is a promising cytotoxic molecule against Follicular lymphoma B cells, while it does not affect normal B cells. We also showed that the downregulation of c-FLIP(L/s) after the inhibition of NF-κB signaling could restore TRAIL-mediated apoptosis in follicular lymphomas [35]. In this current study, we aimed to assess whether the newly identified FLIP inhibitors could sensitize the extrinsic apoptosis pathway when combined with the death ligand TRAIL (Figure 5). The squamous non-small cell lung cancer (NSCLC) cell line, H1703, either mock-transfected or carrying the long form c-FLIP-(L), were initially treated with or without the TRAIL ligand in order to evaluate the influence of c-FLIP(L) on apoptosis. As expected, the high expression of c-FLIP(L) in the transfected cell line inhibits TRAIL-induced apoptosis compared to H1703 mock-transfected cells (Figure 5A), confirming the anti-apoptotic effect of c-FLIP. Since c-FLIP(L) prevents death receptor-mediated apoptosis, we then examined whether its inhibition could increase the sensitivity of cancer cells to TRAIL, since it was reported that targeting c-FLIP directly or indirectly could overcome apoptosis resistance [36,37]. We first investigated the cytotoxic effect of our nine selected molecules, administered alone at the "appropriate concentration" (chosen from a range of concentrations for each molecule with no cytotoxicity on cells). As shown in Figure 5B, there was no enhancement in cell death when the cells were treated with the inhibitors alone, compared to non-treated cells, either in mock or in FLIP(L) cells. This indicates that these FLIP-inhibitors at selected concentrations were not toxic against cancer cells. In contrast, a significant increase in apoptosis was observed in cells overexpressing c-FLIP(L), after a co-treatment of TRAIL with FLIP-inhibitors, compared to FLIP(L) cells treated with TRAIL alone or FLIP inhibitors alone. Molecules 1, 3, 4, 8 and 9 were the most efficient, and showed a remarkable enhancement of cell death compared to other tested molecules ( % of dead cells by apoptosis ≥ 30%). These findings strongly suggest that blocking c-FLIP with these new molecules overcomes c-FLIP resistance and sensitizes cancer cells to death receptor-mediated apoptosis.

### 3.5. A Combination Treatment of TRAIL with c-FLIP-Targeting Molecules Restores Caspases-Dependent Apoptosis

Caspases are the major effectors of apoptosis. Death signaling events induce the dimerization and auto-processing of the initiator caspase-8, which activates downstream executioner caspases such as caspase-3 [38]. PARP cleavage is reported as a hallmark of caspase-3 activation and apoptosis [39]. At high concentrations, c-FLIP(L) forms a heterodimer with caspase-8, which prevents the formation of a fully active caspase-8, thereby blocking apoptosis [40]. To study the impact of the newly identified c-FLIP inhibitors on caspase activation and apoptosis induction, H1703 overexpressing c-FLIP(L) was treated with TRAIL combined with each c-FLIP inhibitor. Among the nine inhibitors, we have chosen only those that showed an inhibitory effect both in cellulo (Figures 4 and 5) and in the pull-down assay (Figure 3), such as Molecules 1, 3, 4 and 9. H1703–c-FLIP(L) cells treated with 100 ng/mL of TRAIL alone did not exhibit any caspase cleavage, confirming the role of c-FLIP in the inhibition of caspase-8/10 activation (Figure 6A, left panels). However, a combination of TRAIL with the selected molecules targeting c-FLIP restored apoptosis, demonstrated by caspase-8, -3 and PARP cleavage. Interestingly, treated cells with c-FLIP inhibitors alone did not induce any caspase cleavage, confirming that these molecules do not exhibit a direct cytotoxic effect on cells (Figure 6A, right panels). Similarly, H1703 lacking c-FLIP(L) protein expression was treated either with TRAIL alone, inhibitors alone or TRAIL combined with c-FLIP inhibitors. In the absence of c-FLIP (L), the cells were sensitive to TRAIL and induced caspase-8 cleavage, as well as subsequently caspase-3 and PARP cleavage. A combination of TRAIL with c-FLIP inhibitors also induced caspase cleavage (Figure 6B, left panels), suggesting that only TRAIL is responsible for caspase activation, as cells treated with the molecules alone did not exhibit any caspase cleavage (Figure 6B, right panels). Taken together, these experiments indicate that the newly charac-

terized inhibitory molecules prevent c-FLIP binding to the FADD and its recruitment into the DISC. Consequently, apoptosis can be restored though efficient caspase-8/10 cleavage.

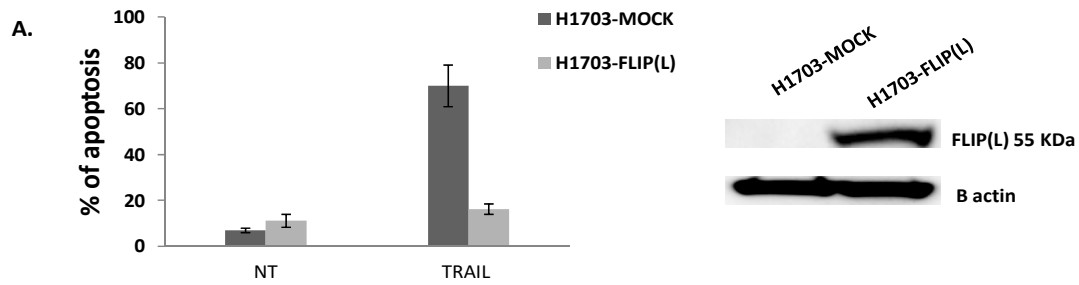

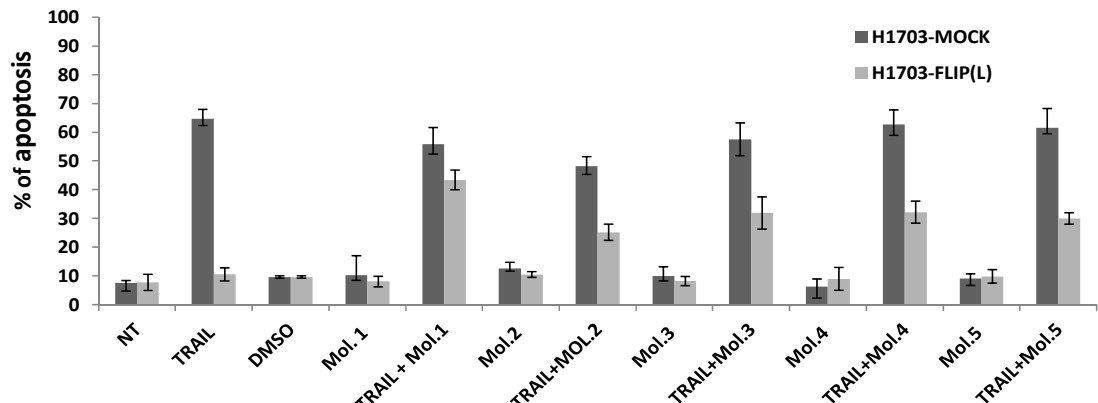

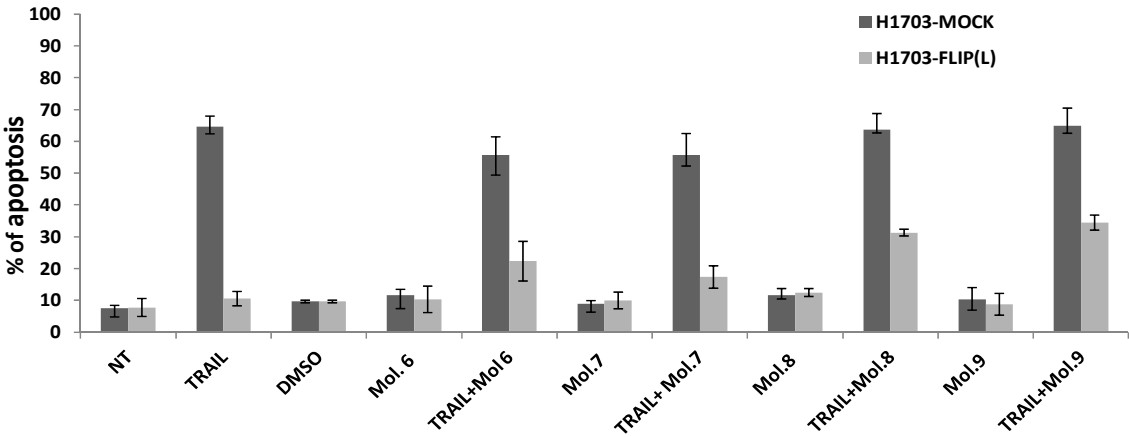

**Figure 5.** Inhibition of c-FLIP rescues TRAIL-mediated apoptosis. A total of (**A**) $2 \times 10^5$ H1703-mock-transfected and H1703-FLIP (L) cells were seeded and treated with or without 100 ng/mL TRAIL for 18 h. Apoptosis was evaluated with Annexin V-PE staining by flow cytometry (*n* = 3). A total of (**B**) $2 \times 10^5$ H1703 mock-transfected and H1703-FLIP (L) cells were seeded and treated with or without 100 ng/mL TRAIL alone, inhibitors of FLIP(L) alone or with co-treatment of TRAIL/FLIP inhibitors for 18 h. Apoptosis was evaluated with Annexin V-PE staining by flow cytometry (Student's *t*-test, *n* = 3).

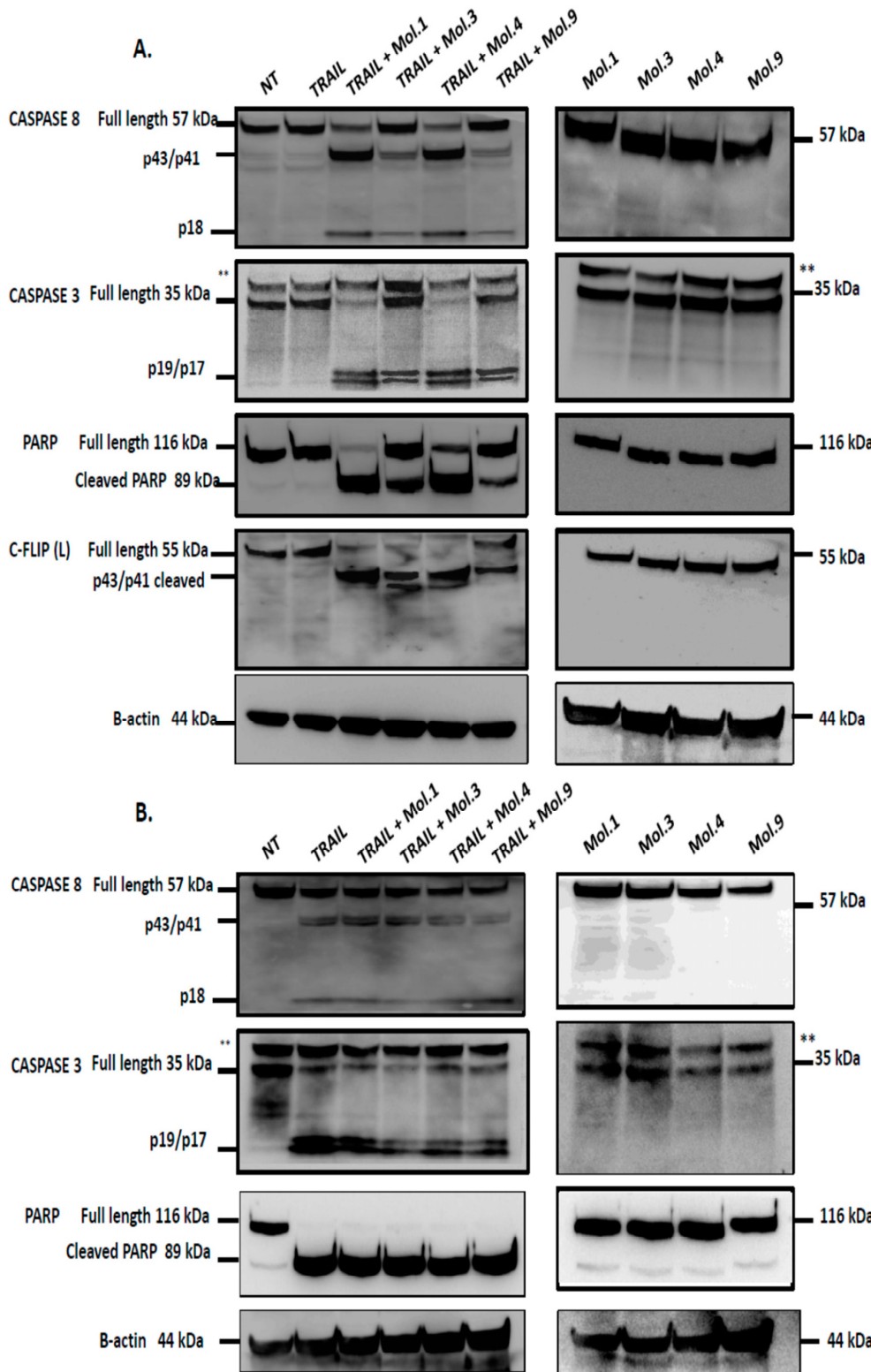

**Figure 6.** Inhibition of c-FLIP restores apoptosis mediated by caspases activation. (**A**) H1703 over-expressing c-FLIP (L), and (**B**) H1703-lacking c-FLIP, (L) were treated with 100 ng/mL of TRAIL alone, 500 μM of c-FLIP inhibitors alone, or TRAIL combined with inhibitor 1, 3, 4 or 9 at their appropriate concentrations for 8 h. Lysates were separated by SDS-PAGE and analyzed by Western Blot using specific antibodies. The bands marked with an asterisk are unspecific bands for the anti-caspase-3 (8G10).

## 4. Discussion

TRAIL's ability to specifically kill tumor cells in vitro and in vivo makes this death ligand or its death receptors a promising anticancer agent or target [41]. However, a number of primary cancer cells are resistant to TRAIL monotherapy. This resistance can occur at the death receptor level via the upregulation of the anti-apoptotic protein c-FLIP [42]. c-FLIP is also a potent resistance factor against other TNFα superfamily members and chemotherapeutic drug-mediated apoptosis [43]. c-FLIP has been found to be overexpressed in many types of malignancies, and its overexpression is associated with poor prognosis and tumor progression, due to the inhibition of the apoptotic process [44]. The three known variants of c-FLIP can interfere with the FADD/caspase-8 interaction, thus inhibiting caspase-8 recruitment into the DISC and blocking its activation, which ultimately leads to the inhibition of apoptosis [14]. Because c-FLIP upregulation prevents apoptosis and leads to cancer promotion, its silencing has been shown to restore cell death. Therefore, c-FLIP is considered an important promising target in cancer therapies. In this context, we aimed to identify new inhibitory molecules specifically targeting c-FLIP and combine them with TRAIL in order to restore apoptosis in cancer cells. It was previously reported in several studies that a combination of targeted anticancer therapies with TRAIL sensitizes resistant cells to TRAIL-induced apoptosis [45,46].

Because of the high structural homology to caspase-8 [14], c-FLIP is a difficult protein to target, since small molecules able to block its recruitment into the DISC may also inhibit caspase-8 recruitment. Therefore, small molecules selectively targeting c-FLIP without affecting caspase-8 functions are required. The challenge of our work was to identify new molecules that can selectively bind to c-FLIP and inhibit its interaction with the FADD within the DISC, without affecting caspase-8 binding. We aimed to construct in silico DED2s of c-FLIP and caspase-8 based on homology models as their human crystallographic structures were not established. Searching for similar target sequences, Yang and collaborators [47] showed that v-FLIP DED2 and FADD DED were structurally similar to DED2 of c-FLIP and caspase-8, respectively. After having successfully generated a 3D-model homology domain of c-FLIP and caspase-8 DED2 and identified a potential unique druggable pocket in c-FLIP (Figures 1 and 2), we initiated in silico screening and docking experiments of 1880 compounds from the NCI database, targeting the homology structures of c-FLIP and caspase-8. This large set of chemical molecules led us to select compounds that only bound to the DED2 of c-FLIP and not of caspase-8.

Among the DED2–c-FLIP binding molecules, we selected nine compounds exhibiting the most potent binding affinity toward the DED2 of c-FLIP, versus the DED2 of caspase-8. Further analyses were required to assess whether these new molecules were able to inhibit DED–FADD/DED2–cFLIP interactions. The FADD is considered the nucleus of the DISC assembly, and responsible for initial caspase-8/-10 and c-FLIP recruitment through homotypic DEDs' interactions [48]. The FADD is composed of two distinct domains: the DD (Death Domain) responsible for receptor engagement, and the DED, which contains a face-exposed hydrophobic patch, conserved in all other DEDs' proteins, and is thought to be crucial for DED–DED interactions with caspase-8 and c-FLIP [49]. Therefore, to elucidate the effect of the nine selected molecules, we investigated their ability to disrupt the interaction between the recombinant human FADD and the full length c-FLIP(S). The long form c-FLIP(L) was indeed not possible to produce and purify, due to its rapid precipitation and low stability. Here, we showed that Molecules 1, 2, 3, 4 and 9 were able to inhibit FADD/c-FLIP(S) interaction in a pull-down assay (Figure 3A–D,I). In addition, Molecules 1, 3, 4 and 9 were able to prevent FLIP recruitment into the DISC using an immunoprecipitation assay, confirming our previous results.

We further assessed whether these nine molecules could inhibit c-FLIP binding to the FADD in a cellular model. It has been reported that c-FLIP is highly expressed in non-small cell lung cancer (NSCLC), where its high cytoplasmic expression correlates with poor prognosis. Moreover, c-FLIP inhibits anticancer drug-induced apoptosis in preclinical models of NSCLC [50]. Thus, we overexpressed the long form of the c-FLIP (L) protein in

H1703 cancer cells (NSCLC), and we treated these cells first with TRAIL alone. As observed in Figure 5A, an ectopic expression of c-FLIP (L) inhibits TRAIL-mediated apoptosis. In contrast, when c-FLIP (L) is absent, cancer cells were sensitive to TRAIL, confirming the primary function of c-FLIP as an inhibitor of apoptosis and a possible biomarker of tumor resistance. As the activity of TRAIL is prevented by c-FLIP overexpression, targeting c-FLIP using our new molecules may restore TRAIL function, as it is broadly referenced in several studies showing that c-FLIP silencing sensitizes tumor cells to death ligand-induced apoptosis [51]. To evaluate the efficiency of our newly selected molecules in inhibiting c-FLIP function, a combination of them with TRAIL was assessed, and showed a remarkable enhancement in the apoptotic level in c-FLIP-overexpressing cells; meanwhile, the same molecules exhibit no cell death when administered alone, compared to non-treated cells (Figure 5B). These findings revealed that the newly identified molecules are distinctly efficient against c-FLIP and help to restore apoptosis in TRAIL-resistant cells, while they are not cytotoxic alone.

Molecules 1, 3, 4 and 9 are able to restore apoptosis when combined with TRAIL (Figure 5B), confirming their binding affinity to c-FLIP (Figure 3) and their role in preventing c-FLIP recruitment into the DISC (Figure 4). Previous studies have shown that these molecules target and inhibit the following enzymes: protein-tyrosine phosphatase (for Molecule 1), MAP kinase (Molecules 3 and 4) and matrix metalloproteinase (Molecule 9). In contrast, while Molecules 2 and 6 showed an inhibitory role in the FADD/c-FLIP interaction in the molecular assay (Figure 3B,F), they poorly restored apoptosis in resistant cancer cells overexpressing c-FLIP (Figure 5). Such results indicate that these two molecules may lose their binding potential to c-FLIP within the cell, and they might require further chemical and structural modifications in order to enhance their binding affinity to c-FLIP. Surprisingly, we found that Molecules 5, 7 and 8 could not prevent the FADD/c-FLIP interaction in the pull-down assay (Figure 3E,G,H). In contrast, they were able to sensitize resistant cells to TRAIL-induced apoptosis, especially Molecule 8, which enhanced apoptosis by more than 30%, with no cytotoxicity by itself. This evidence led us to conclude that these three molecules are not selective to c-FLIP, and that they may target other signaling pathways contributing to cell death.

c-FLIP competes with caspase-8 recruitment for the DISC and prevents its activation, thereby blocking downstream caspases' activation and apoptotic pathway [52]. The downregulation of c-FLIP using siRNAs sensitizes cells to caspases-dependent apoptosis [53]. In this study, we investigated caspases' activity after a combination of TRAIL with c-FLIP-inhibitory molecules to evaluate the restoration of extrinsic apoptosis. Our data indicates that caspase activity is blocked when c-FLIP is active and recruited into the DISC (Figure 6A). However, inhibiting c-FLIP function with these new molecules enhances caspases-8, -3 and PARP cleavages, and promotes TRAIL-mediated apoptosis. Moreover, when cells overexpressing c-FLIP are treated with the molecules alone, no caspases activation and PARP cleavage have been observed, indicating that the administration of these molecules is safe and does not induce any cytotoxicity. Similarly, when cells lacking c-FLIP (L) were treated with these compounds alone, we did not observe any caspase cleavage, suggesting that these molecules exhibit no side effect on cells with no expression of c-FLIP.

In conclusion, we report that c-FLIP expression is a relevant biomarker of cancer resistance in general, and for the anticancer agent TRAIL. In this current study, we have demonstrated that c-FLIP function can be inhibited by new small molecules, while caspase-8 remains unaffected. The inhibition of c-FLIP interactions with the FADD precludes c-FLIP from its recruitment into the DISC, and allows caspase-8's binding to the FADD, promoting TRAIL-induced apoptosis in resistant cancer cells. This combination therapy of TRAIL with these c-FLIP-targeted new agents could be a promising approach to eradicate tumors with c-FLIP upregulation. However, maximizing the binding affinity of the most efficient molecules through structural modifications is required, in order to restore a higher level of apoptosis in tumor cells.

**Author Contributions:** Software, S.A.-S. and P.B.; Validation, R.D. and T.G.; Investigation, K.Y.; Resources, R.P., P.L. and U.J.; Writing original draft, K.Y.; Writing—review & editing, T.G.; Visualization, K.Y.; Supervision, T.G. All authors have read and agreed to the published version of the manuscript.

**Funding:** This research received no external funding.

**Institutional Review Board Statement:** Not applicable.

**Informed Consent Statement:** Not applicable.

**Data Availability Statement:** Data are contained within the article.

**Conflicts of Interest:** The authors declare no conflicts of interest.

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
