# Peer review of "The Identification of New c-FLIP Inhibitors for Restoring Apoptosis in TRAIL-Resistant Cancer Cells"

_cimb, doi:10.3390/cimb46010046_

Round 1

Reviewer 1 Report

Comments and Suggestions for Authors

The present manuscript entitled "Identification of New c-FLIP Inhibitors to Restore Apoptosis in TRAIL-Resistant Cancer Cells" shows a compelling exploration into the development of small inhibitory molecules targeting c-FLIP, a caspase-8 homologous protein with antiapoptotic properties in various cancer types. The study employs an in-silico approach, utilizing a homology 3D model of c-FLIP and screening compounds from the NCI database, to identify potential candidates for inhibiting c-FLIP and restoring apoptosis in TRAIL-resistant cancer cells. The study suggests a promising strategy to overcome immune and anticancer drug resistance by targeting c-FLIP and restoring caspase-8 dependent apoptotic cascade.

In general, the manuscript is well written and concise. However, before it can be accepted for publication, some points must be clarified, and authors need to address the following main issues:

1) In the introduction, the authors mention c-FLIP inhibitors, such as cisplatin and actinomycin D, and emphasise how siRNA-based approaches open up promising avenues in contrast to traditional methods. To enhance this discussion, I suggest expanding the spectrum of c-FLIP inhibitors. Are there peptide-based inhibitors? Their activity could demostrate more selective see doi 10.2174/09298673113209990184, 10.2174/0929867311320140006.

2) The study highlights differences in the action of various molecules, particularly noting that molecules 1, 3, 4, and 9 inhibit FADD-c-FLIP interaction, whereas molecules 2 and 6, although inhibitory in molecular assays, do not restore apoptosis in resistant cancer cells overexpressing c-FLIP. Additionally, molecules 5, 7, and 8, which did not prevent FADD/c-FLIP interaction in the pull-down assay, still sensitized cells to TRAIL-induced apoptosis.

3)  Can the authors provide insights into whether these observed differences in action are attributed to variations in the chemical structure of the molecules? Moreover, are there specific structural features or modifications that may contribute to the observed functional differences? Addressing this aspect could provide valuable insights into the molecular mechanisms underlying the specificity and efficacy of the identified compounds.

3) Since molecules 5, 7 and 8 did not prevent the FADD/c-FLIP interaction in the pull-down assay but still sensitised cells to TRAIL-induced apoptosis, are the authors able to provide more information on potential off-target effects and alternative signalling pathways influenced by these molecules?

MINOR REVISIONS

1) The authors should uniform the terminology used to refer to the molecules studied (1-9) both in the text and in the figures, choosing 'compounds' or 'molecules' for consistency.

2) Please uniform references in the text: line 62 (Chang et al., 2006; Golks 62 et al., 2005) and line 77 (Scaffidi et al., 77 1999)

3) In my opinion, for enhanced clarity and better reader comprehension, it would be more suitable to assign horizontal sequential numbering to the figures, rather than the vertical numbering currently used in Figure 1. Additionally, if feasible, improving the quality of the figures would further contribute to the overall visual presentation.

4) Please align all panels in Figure 3 with each other, paying particular attention to panels C and G, to enhance overall consistency and visual clarity.

Comments on the Quality of English Language

Moderate revision of spelling and building of sentence

Author Response

Reply to the main issues:

1) There is no peptide-based inhibitors to FLIP Protein

2) The selected molecules are very diverse structurally, it is therefore not possible to establish a structure-activity relationship (SAR) for these molecules. To do SAR, it would be necessary to select analogues of the active ingredients and test them, but this is not the subject of this publication.

3)  as these molecules were not used in previous studies, little informations are known regarding their effects in cells. However, we could only knew that molecules 5, 7 and 8 could bind to the anti-apoptotic protein Mcl1 and sequester its activity. For molecules 2 and 6, no informations in litterature but we suggest that they are unstable and lose their binding affinity inside the cells.

Minor Revisions answers:

1) we will use the term "Molecule"

2) we have corrected the references

3) We put the figures in a vertical way due to their different dhapes so they fit better in WORD sheet

4) Figure 3 panels adjusted

Reviewer 2 Report

Comments and Suggestions for Authors
  • A brief summary
    The aim of the work was the identification and characterization of cFLIP inhibitors. cFLIP is a protein overexpressed in many types of cancers. Its activity prevents cellular apoptosis and makes cancer cells resistant to chemotherapy. Thus, including cFLIP inhibitors in chemotherapy should improve the effective elimination of cancer cells. The Authors selected nine compounds as potential cFLIP inhibitors from 1882 compounds using molecular modeling methods and then tested their impact on cFLIP-FADD interactions in vitro using recombinant proteins. Observations were verified on the cancer cell model with overexpression cFLIP. The Authors have shown that selected compounds were very effective in the TRAIL-mediated apoptosis in the presence of cFLIP. Therefore, these compounds can be used in further treatment of cFLIP-positive cancer.
  • General concept comments
    The work is well-designed. Experiments are adequate and performed properly with required controls. Results are presented clearly, so I did not find any severe inaccuracies.
  • Specific comments
    - Line 62 -- no uniform cited way- the numbers 15 and 16 should be instead of names Chang and Golks,
    - Line 77-- no uniform cited way- it should be 20 )
    - Line 229 -- the wrong aim- it should be “to prepare cellular lysates” rather than “to evaluate enzymatic activity”.
    - Line 292 -- the pull-down assay is not performed “in cellular”, so please remove this information.

    - Line 401 -- Western Blot instead of western Blot

General questions research articles:

  • Is the manuscript clear, relevant for the field and presented in a well-structured manner? Yes
  • Are the cited references mostly recent publications (within the last 5 years) and relevant?
    Only 4 references were published within the last 5 years, the rest (49 before that time).

Does it include an excessive number of self-citations? No

  • Is the manuscript scientifically sound and is the experimental design appropriate to test the hypothesis? Yes
  • Are the manuscript’s results reproducible based on the details given in the methods section? Yes
  • Are the figures/tables/images/schemes appropriate? Yes

Do they properly show the data? Yes. However, in case Figure 5, the statistical analysis should be included.

Are they easy to interpret and understand? Yes, it is very clear.

Is the data interpreted appropriately and consistently throughout the manuscript? Yes

Please include details regarding the statistical analysis or data acquired from specific databases.

  • Are the conclusions consistent with the evidence and arguments presented? Yes
  • Please evaluate the ethics statements and data availability statements to ensure they are adequate.
    The work is not performed on animals or human tissues. All statements are declared correctly.
  • Novelty: Is the question original and well-defined?
    The problem is not new, but the way of resolving it is novel.
    Do the results provide an advancement of the current knowledge?
    Yes
  • Scope: Does the work fit the journal scope?
    Yes, (Molecular Cell Biology, Molecular Oncology, Molecular Pharmacology)
  • Significance: Are the results interpreted appropriately? Yes
    Are they significant?
    Yes
    Are all conclusions justified and supported by the results? Yes
    Are hypotheses carefully identified as such? Yes
  • Quality: Is the article written in an appropriate way? Yes
    Are the data and analyses presented appropriately?
    Yes
    Are the highest standards for presentation of the results used?
    Yes
  • Scientific Soundness: Is the study correctly designed and technically sound? Yes
    Are the analyses performed with the highest technical standards?
    Generally yes. However, there is a lack of presentation of statistical analysis in Figure 5.
    Is the data robust enough to draw conclusions?
    Yes
    Are the methods, tools, software, and reagents described with sufficient details to allow another researcher to reproduce the results?
    Generally yes. However, obtaining the cancer cell line with overexpression of CFLIP could be more detailed.
    Is the raw data available and correct (where applicable)?
    Not all raw data are available.
  • Interest to the Readers: Are the conclusions interesting for the readership of the journal? Yes
    Will the paper attract a wide readership, or be of interest only to a limited number of people?
    As I have mentioned previously, the work should be interesting for scientists working on cancer cell biology, pharmacologists, and clinicians.
  • Overall Merit: Is there an overall benefit to publishing this work? Yes
    Does the work advance the current knowledge?
    Yes
    Do the authors address an important long-standing question with smart experiments? Yes
    Do the authors present a negative result of a valid scientific hypothesis? No
  • English Level: Is the English language appropriate and understandable? Yes

Author Response

In the figure 5, A student t test was done as a statistical analysis. all of the results were from at least three experiments.

The H1703 cell line with FLIP overexpression was provided by another research group. We did not proceed the transfection in our lab.

Reviewer 3 Report

Comments and Suggestions for Authors

The manuscript reported by Yaacoub et al. presents a rational design of new molecules targeting c-FLIP inhibition.

The authors support their hypothesis correctly in the introduction, although this part is perhaps a bit long. In the materials and methods section, the authors detail all the procedures and then present the results. The final discussion also seems too long and does not provide as much information as it should. The authors limit themselves to explaining their results again.

In view of the manuscript, this reviewer requests that the authors make modifications and provide more information following these points: 

-The introduction has too much information. The authors should summarise the information and also provide an explanatory figure to illustrate their main idea. 

-More information is needed on how the nine compounds in figure 1B were selected. This reviewer does not find an explanation in the text. Neither in the methodology part nor in the results (in fact, figure 1B is not cited in the text).

-Regarding these compounds, some structures are deformed and the bonds are not well drawn. This reviewer wonders whether the authors want to put the "conformation" they adopt on the target. If so, this is not the right way. If this is not the case, the authors' explanation is awaited. 

-It would also be useful to have a more extensive explanation of how the compounds interact on the target (hydrogen bonds, aa with which they interact). The information on this part is very poor. 

-There are also no references related to the synthesis of the compounds. 

-The authors present numerous Western Blots. This reviewer considers it necessary to provide the original images indicating the loading control and the molecular weight markers used. 

-Regarding the apoptosis studies, it would be useful to add to the supplementary material the dot plots at least of the control and those conditions where the most outstanding results have been obtained.

Comments on the Quality of English Language

With regard to the quality of the English language, authors need to review formal aspects such as the use of italics in expressions like "in silico" or to be careful with unit expressions. 

Author Response

1)More information is needed on how the nine compounds in figure 1B were selected:

Answer: chemical libraries to be used for virtual screening. From free virtual chemical libraries whose molecules are immediately available, such as the NCI, ZINC, e-MOLECULES as well as ICOA’s own chemical library, we opted for the 1880 compounds from the NCI DiversitySet3 extracted from the ZINC database.

Methodolgy: I have added it in the text. We used three different docking softwares to direct the 1880 molecules toward the druggable pockets of FLIP and casp8. Molecules bound on casp8 are not selective. we chose only that bound on FLIP. The top 3 molecules of each software classification were selected as they are considered selective to FLIP and have the highest binding affinity to its pocket. The selection of molecules is well specified in the manuscript (5) Hit selection and in the text

2) concerning the best poses of the 9 compounds in the c-FLIP model : there is no formation of hydrogen bonds because the interactions are of the aromatic/hydrophobic type. The residues involved are notably Tyr119 at the entrance to the binding pocket, which most of the time stacks with an aromatic core of the molecule, and 3 leucines in the pocket, Leu112, Leu138 and Leu15

2) There are also no references related to the synthesis of the compounds.

Answer: the nine molecules were the only ones easily obtained without synthesis

3) We have provided the original images of Western Blot to Mrs Linnie YINN

4) English revisions are done as requested

Round 2

Reviewer 1 Report

Comments and Suggestions for Authors

I am satisfied with authors' modifications

Comments on the Quality of English Language

It is well written

Author Response

Thank You very Much

Reviewer 3 Report

Comments and Suggestions for Authors

The authors have addressed most of the improvements proposed by this reviewer.

However, Figure 1B is still lacking in quality and formatting. I strongly encourage authors to correct this figure to improve their manuscript.

Author Response

Figure 1B is well adjusted. we really did everything possible to clearly show the structures of the molecules with their numbers in a single figure.

Molecules are now shown in a horizontal panel as it was requested before